# A Comprehensive Technology Platform for the Rapid Discovery of Peptide Inhibitors against SARS-CoV-2 Pseudovirus Infection

**DOI:** 10.3390/ijms241512146

**Published:** 2023-07-29

**Authors:** Marten Beeg, Sara Baroni, Arianna Piotti, Alessia Porta, Ada De Luigi, Alfredo Cagnotto, Marco Gobbi, Luisa Diomede, Mario Salmona

**Affiliations:** Department of Molecular Biochemistry and Pharmacology, Istituto di Ricerche Farmacologiche Mario Negri IRCCS, Via Mario Negri 2, 20156 Milan, Italy; sara.baroni@marionegri.it (S.B.); piottiarianna@gmail.com (A.P.); alessia.porta@marionegri.it (A.P.); ada.deluigi@marionegri.it (A.D.L.); alfredo.cagnotto@marionegri.it (A.C.); marco.gobbi@marionegri.it (M.G.); luisa.diomede@marionegri.it (L.D.); mario.salmona@marionegri.it (M.S.)

**Keywords:** drug discovery pipeline, COVID-19, SARS-CoV-2, peptide inhibitors, macrocyclic peptides, ELISA and SPR assays, pseudovirus entry assay

## Abstract

We developed and validated a technology platform for designing and testing peptides inhibiting the infectivity of SARS-CoV-2 spike protein-based pseudoviruses. This platform integrates target evaluation, in silico inhibitor design, peptide synthesis, and efficacy screening. We generated a cyclic peptide library derived from the receptor-binding domain (RBD) of the SARS-CoV-2 spike protein and the angiotensin-converting enzyme 2 (ACE2) receptor. The cell-free validation process by ELISA competition assays and Surface Plasmon Resonance (SPR) studies revealed that the cyclic peptide c9_05, but not its linear form, binds well to ACE2. Moreover, it effectively inhibited the transduction in HEK293, stably expressing the human ACE2 receptor of pseudovirus particles displaying the SARS-CoV-2 spike in the Wuhan or UK variants. However, the inhibitory efficacy of c9_05 was negligible against the Omicron variant, and it failed to impede the entry of pseudoviruses carrying the B.1.351 (South African) spike. These variants contain three or more mutations known to increase affinity to ACE2. This suggests further refinement is needed for potential SARS-CoV-2 inhibition. Our study hints at a promising approach to develop inhibitors targeting viral infectivity receptors, including SARS-CoV-2’s. This platform also promises swift identification and evaluation of inhibitors for other emergent viruses.

## 1. Introduction

The coronavirus disease 2019 (COVID-19) was first identified in December 2019 and rapidly spread worldwide, infecting more than 750 million people. The COVID-19 pandemic has caused a significant global impact, claiming the lives of over 6.8 million individuals [1]. The disease is caused by the severe acute respiratory syndrome coronavirus 2 (SARS-CoV-2), which shares close to 80% of its genome sequence with the severe acute respiratory syndrome coronavirus (SARS-CoV-1) [2]. SARS-CoV-2, SARS-CoV-1, and MERS-CoV are all part of the genus beta coronavirus, composed of single-stranded RNA viruses. These viruses infect the lower respiratory tract of humans, causing a lung infection called pneumonia [3].

The COVID-19 pandemic has created an unprecedented global public health crisis, leading to an urgent need to develop effective drugs to combat the virus. To address this need, developing a well-structured drug discovery pipeline is crucial. Accurate target identification is the crucial first step in the drug discovery pipeline. Once the target is identified, the next step is proposing new molecules via screening or rational design. To test the effectiveness of the proposed molecules, high-throughput assays such as ELISA can be employed. ELISA is a rapid, cost-effective, and sensitive technique that can detect the binding of a small molecule to its target protein. This approach enables quick identification of the most promising molecules and can be further refined through techniques testing for compound stability, specificity, and affinity. Finally, a highly flexible pseudovirus platform can be employed to test the efficacy of selected compounds on cellular systems. Pseudoviruses are non-infectious virus particles containing the same surface proteins as wild-type viruses. Due to the possibility of handling the pseudoviral systems in intermediate biosafety levels (BSL2), they can be used to evaluate the neutralizing activity of the selected compounds against different viral strains, including emerging variants. 

The SARS-CoV-1 and SARS-CoV-2 viruses use the spike protein (S-protein) to bind to the angiotensin-converting enzyme 2 (ACE2) on host cells, enabling them to enter and infect the cells [4]. ACE2 is crucial in regulating blood pressure, diabetes, and maintaining heart function [5]. It is expressed in various organs, including the lung, heart, kidney, small intestine, stomach, and oral mucosa, with the lung being the most vulnerable organ to SARS-CoV-2 due to its extensive surface area [6]. The S-protein of SARS-CoV viruses consists of two subunits, the S1 subunit, which binds to host receptors, and the S2 subunit, which facilitates the fusion of viral and cellular membranes [7]. The receptor-binding domain (RBD) of the S1 subunit binds to the extracellular peptidase domain of ACE2, and upon binding, the S protein is cleaved at the S1/S2 site by the transmembrane serine protease 2, allowing the S2 subunit to facilitate viral entry into host cells [8]. The binding affinity between the receptor-binding domain (RBD) variants of the SARS-CoV-2 virus and ACE2 can differ significantly. The RBD of SARS-CoV-2 exhibits a higher binding affinity to human ACE2 than the RBD of SARS-CoV-1. Furthermore, the Omicron variant demonstrates an enhanced affinity for human ACE2 compared to the original SARS-CoV-2 RBD [7,9,10]. Moreover, some RBD variants, like the Omicron BA.1 variant, have mutations that allow them to escape convalescent sera and monoclonal antibodies that are effective against earlier virus strains [11]. 

Due to its critical role in viral entry, the S-protein/ACE2 interaction is a primary therapeutic target for treating SARS-CoV-2 infections [12]. Various approaches have been developed, such as engineered ACE2 mimetics or decoy receptors [13], de novo-designed mini proteins [14], macrocyclic peptides [15,16], and synthetic peptide libraries [17] that are capable of binding to the SARS-CoV-2 spike RBD. Another strategy is the development of anti-ACE2 inhibitors to block the interaction between RBD and ACE2 [18]. However, since ACE2 has essential functions in the body, anti-ACE2 inhibitors should not interfere with its physiological processes. A small and highly specific inhibitor, such as a small peptide, would be an ideal solution to block SARS-CoV-2 infection. Peptides are cost-effective, easy to develop, and have a lower risk of toxicity and side effects than other drug classes. Macrocyclic peptides have been shown to be effective in disrupting protein–protein interactions [19,20,21], specifically where a defined binding pocket is missing, such as in the ACE2-RBD interaction [22].

Here, we propose a drug discovery pipeline that employs ELISA, Surface Plasmon Resonance (SPR), and pseudovirus-based cellular assays to test the efficacy of rationally designed macrocyclic peptides. Our focus is identifying peptides that bind to ACE2, inhibiting the ACE2-RBD interaction and blocking viral entry into host cells.

## 2. Results

### 2.1. Drug Discovery Pipeline

We designed and developed a drug discovery pipeline, as shown in Figure 1, to efficiently and effectively identify potential drug candidates for COVID-19. 

The pipeline begins with target evaluation, where we gather genetic and structural information on the target. This information is then used in the inhibitor identification step, where we may utilize in silico tools to design molecules based on the previous step or utilize compounds suggested by the literature or external partners.

If our partners provide compounds, they bypass the inhibitor synthesis phase and move directly into the validation process. Otherwise, compounds designed by our team will be synthesized and then screened for efficacy in the validation process. The validation process consists of two steps: a cell-free process and an in vitro approach.

The cell-free process includes ELISA and SPR assays, which test the direct interaction between the compounds and the target or the competition of the inhibitor with the target. This step helps to identify promising compounds for further testing. The compounds that pass this stage are then moved to the in vitro process, which incorporates a pseudovirus entry assay. This assay is designed to test the ability of compounds to inhibit viral entry into cells, a critical step in the pathogenesis of COVID-19. Our drug discovery pipeline is designed to be efficient and effective, enabling us to identify potential drug candidates for COVID-19 rapidly.

### 2.2. RBD-Derived Binder and Short Library Creation

We employed the pipeline to design and test peptides potentially effective against SARS-CoV-2 infection. We used the Rosetta Peptiderive protocol to extract small peptide binders/inhibitors based on Wuhan-type RBD from the ACE2/RBD binding interface (Figure 2A). The protocol yielded a cyclizable peptide of nine amino acids flanked by two cysteines (c9 peptide), exhibiting significant free binding energy (Figure 2B). This peptide spans the amino acid sequence 498 to 506 of RBD, including the crucial N501Y mutation. We performed single and two-point mutations at the hot spot positions identified by Starr et al. [23] to create a library of 11 sequences (c9_01 up to c9_10), which includes the c9 peptide in the wild-type form (Figure 2C). Our hypothesis suggests that these mutations, which enhance the binding affinity between RBD and ACE2, may also elevate the affinity of the small peptides.

### 2.3. Screening the Library for Potential Inhibitors by ELISA

We conducted an ELISA competition assay using the Wuhan-type RBD protein and ACE2 receptor to identify potential inhibitors from the generated peptide library. The ELISA competition assay was used to evaluate the percentage of inhibition of RBD binding to ACE2 for all cyclizable peptide variants (c9_01 to c9_10) at 100 μM. Pre-incubation of ACE2 with the Wuhan-type RBD as positive control and the peptide variants was performed before adding 25 μM RBD. Notably, the positive control inhibited the RBD binding by 95% at 1 µM, indicating its high efficacy. Furthermore, the z-factor of the screening assay was 1, demonstrating the robustness and reliability of the assay. The results are summarized in Appendix A. The c9 wild-type peptide was not active, while the other variants showed varying levels of inhibition ranging from 0% up to 83%. Variants c9_02 and c9_05 exhibited the highest inhibition at 83% and 78%, respectively, while variants c9_06 and c9_08 showed no inhibition. Variants c9_03 and c9_04 also demonstrated relatively high inhibition at 71% and 33%, respectively. Compounds that exhibited a percent inhibition greater than 50% (c9_02, c9_03, and c9_05) were selected to evaluate their binding affinity using SPR.

### 2.4. SPR Assay for Determining the Binding Affinity of Selected Peptide Variants

We utilized a SPR assay to determine the binding affinity of selected peptide variants with the ACE2 protein bound to the chip surface using as a positive control Wuhan-type RBD. SPR experiments confirmed a RBD binding affinity (K_D_) value of 9.21 nM, comparable to that reported in the literature (K_D_ = 6.34 nM) (Appendix A) [24]. Different peptide concentrations (3, 10, 30, and 100 µM) in PBST were injected in parallel for 180 s with ACE2 bound to the chip surface. The association phase was followed by a 600 s long dissociation phase, where only the buffer flowed. Among the forms tested, only cyclic, but not linear, c9_05 showed a specific binding signal (Figure 3A,B). At the same time, c9_02 did not bind at all, and c9_03 showed high, non-specific binding on the control channel (Appendix A). Although the binding signal of c9_05 is complex and cannot be fitted perfectly with a one-by-one binding model, the estimated K_D_ value using a 1:1 model is 19.9 µM. This is consistent with the ELISA competition assay, which gave 7.5 µM (Figure 3C). The linear form of c9_05 did not compete in the ELISA assay (Figure 3D), supporting the cyclic variant’s specificity. 

### 2.5. Peptide c9_05 Inhibits SARS-CoV-2 Pseudovirus Entry into HEK293-ACE2 Cells

Before testing the selected peptide variants in the pseudovirus entry assay [25], we first confirmed the efficacy of the assay. We ensured that we could obtain a specific signal by comparing the transduction efficiency of pseudovirus-exposing SARS-CoV-2 Wuhan or B.1.351 SA spike variants in HEK293 cells with or without ACE2 receptor expression. As shown in Appendix A, the pseudovirus transduction efficiency was significantly higher in HEK293-ACE2 cells stably expressing ACE2 receptor compared to normal HEK293 cells. The results demonstrate that pseudovirus entry efficiency in HEK293 cells without endogenous ACE2 receptor expression was lower than 10%, indicating the specificity of the pseudovirus entry assay and validating its use for testing the selected peptide variants.

To validate the platform, we evaluated the efficacy of the Wuhan and Omicron RBD variants against pseudovirus entry across a concentration range of 3–100 nM. Pre-treatment of HEK293-ACE2 cells with these variants inhibited the entry of pseudovirus particles exhibiting the B.1.1.7 (UK) or B.1.351 (SA) variant spike protein. Specifically, the Wuhan RBD variant demonstrated inhibitory effects against lentiviral infectivity starting from 100 nM, as seen in Appendix A. Conversely, the Omicron RBD variant was effective at lower concentrations, protecting 3 nM against the B.1.351 (SA) variant and at 30 nM against the B.1.1.7 (UK) variant (Appendix A). 

Before proceeding with the pseudovirus entry assay using the cyclic c9_05 peptide, we evaluated its potential toxic effect on HEK293-ACE2 cells. The peptide at concentrations ranging from 1 to 100 μM did not cause any significant decrease in cell viability (Appendix A).

Finally, we incubated HEK293-ACE2 cells with or without 25 μM cyclic c9_05 and transduced them with pseudoviral particles exposing different SARS-CoV-2 spike isoforms. This peptide concentration was chosen because it was close to the K_D_ calculated from SPR experiments. As shown in Figure 4A, cyclic c9_05 significantly reduced pseudoviral transduction for Wuhan B.1.1.7 UK and B.1.1.529 Omicron variants but not for the B.1.351 SA isoform. The representative fluorescence microscopy images in panel A demonstrate the GFP or RFP signal reduction after c9_05 treatment. The percentage of GFP- or RFP-transduced cells quantified in panel B showed a 30% reduction in the entry for both Wuhan and B.1.1.7 UK and 10% for Omicron variants.

## 3. Discussion

This study has developed a drug discovery pipeline specifically tailored to identify potential drug candidates for COVID-19 efficiently and effectively. The pipeline commences with a thorough target evaluation, which involves gathering genetic and structural information to target a specific mechanism of action. The next step is inhibitor identification, which may include using in silico tools to design molecules based on the target evaluation or utilizing compounds recommended by the literature or external partners. The validation process includes cell-free techniques and an in vitro pseudovirus entry test to identify the most promising compounds.

In this study, we created a cyclizable peptide of nine amino acids flanked by two cysteines. The design of this peptide was guided by the sequence within the RBD region, which interacts with ACE2. To explore a wider range of sequence variations, we generated a library of 11 sequences by introducing single and two-point mutations at known hot spot positions. Through an ELISA competition assay, three compounds were identified that exhibited a percent inhibition greater than 50%. These compounds were further evaluated for their binding affinity using SPR. Among the three variants tested, only c9_05 showed a specific binding signal, c9_02 did not bind, and c9_03 showed high, non-specific binding. The estimated Kd value of c9_05 was found to be 19.9 µM, which is consistent with the ELISA competition assay IC50 = 7.5 µM. Additionally, peptide c9_05 inhibited SARS-CoV-2 pseudovirus entry into HEK-ACE2 cells ranging from 10% to 30% at a concentration of 25 µM. 

The identified small inhibitor, c9_05, derived from the RBD sequence (positions 498–506), displayed binding to ACE2 and blocking pseudovirus entry. The ability of c9_05 to inhibit SARS-CoV-2 may stem from the combined effects of two-point mutations at RBD positions 503 and 505 since single-point mutations in the tested small peptides did not prove effective. Interestingly, single-point mutations at these positions in the full-length RBD did not exhibit as much increase in binding affinity as the N501Y single-point mutation found in the UK and SA variants of SARS-CoV-2 [23]. 

However, it is essential to highlight that the inhibitory effectiveness of c9_05 exhibited significant variation across the different SARS-CoV-2 variants. While it demonstrated some potential against the Wuhan and UK variants, its efficacy diminished greatly against highly mutated variants like Omicron and SA variants, with a negligible inhibition at around 10% for the former and complete ineffectiveness against the latter.

Both the Omicron and SA variants share crucial RBD mutations. Specifically, the SA variant carries three distinct mutations (E484K, N501Y, and K417N) that quadruple its ACE2 affinity [26], likely counteracting c9_05’s inhibitory effect. With Omicron harboring these and more RBD mutations, it is highly plausible that c9_05’s effectiveness is markedly diminished against these heavily mutated variants.

Although c9_05 has a relatively high IC50 of 7.5 µM in the blocking assay, a peptide dimer may increase affinity due to avidity, and multimer inhibitors have been previously shown to be more potent. For example, trimerized miniproteins binding to RBD have been effective, and anti-ACE2 peptide dimers have been shown to increase blocking capacity significantly, reducing the IC50 from 3.5 µM to 70 nM [27]. Additionally, incorporating non-canonical amino acids into the peptide sequence could enhance affinity. Thijssen et al. showed that introducing a non-natural amino acid substitution in a macrocyclic peptide inhibitor of the SARS-CoV-2 spike protein improved binding affinity nearly 10-fold [15]. Specific positions in c9_05 could be evaluated for substituting natural amino acids with optimal non-canonical alternatives. Multimerization or linkage of individual c9_05 peptides into a higher-avidity compound could also potentially overcome limitations against highly mutated viral variants. Employing optimization strategies like non-canonical amino acid substitution and multimerization presents opportunities to build upon the current work and develop even more potent SARS-CoV-2-inhibiting cyclic peptides.

The cyclic peptide c9_05 identified in this study represents a good starting point. However, additional investigations are necessary to improve its potency, selectivity, and pharmacokinetics and assess its effectiveness in vivo. Furthermore, peptides may offer several advantages over other protein-based inhibitors, such as antibodies, when fighting pandemics like COVID-19. Compared to the lengthy process of developing monoclonal antibodies, peptides provide the benefits of rapid and efficient production, as well as the ability to be quickly discovered and synthesized on a large scale using in silico or other high-throughput techniques. These features make them a promising tool for combatting pandemics. Moreover, peptides may have low toxicity concerns due to their minimal immunogenicity and the absence of Fc-mediated side effects like antibody-dependent enhancement.

## 4. Materials and Methods

### 4.1. Cyclizable Peptide Identification from RBD/ACE2 Interface

We utilized the peptiderive protocol to identify a cyclizable peptide sequence from the RBD/ACE2 interface based on the RBD protein [28,29]. Peptiderive is a self-contained executable that utilizes the Rosetta software suite (https://www.rosettacommons.org/software, accessed on 7 June 2023) as its fundamental engine for predicting and designing protein–protein interface-derived peptides. It was developed to simplify the process of selecting and assessing peptide sequences that bind to protein interfaces. To identify the cyclizable peptide sequence, we first minimized the complex structure (pdb: 6M0J) using the ref2015 energy function to eliminate any local clashes [30]. Then, we employed a sliding window approach with a size range of 6 to 10 amino acids over the RBD protein chain to assess each sequence’s binding capability and cyclizability at the interface. The software reports positive results that can be used for further studies. For molecular visualization and analysis in this study, we employed the PyMOL Molecular Graphics System, version 2.5 (Schrödinger, LLC, New York, NY, USA) [31].

### 4.2. Solid-State Peptide Synthesis

All peptides were synthesized simultaneously by solid-phase chemistry using fluorenylmethyloxycarbonyl chloride group (Fmoc)-protected amino acid with SyroI peptide synthesizer (Biotage, Uppsala, Sweden). The synthesis was performed at a 0.1 mM scale on Rink amide resin (Burlinton, MA, USA). Fmoc deprotection was performed automatically at room temperature by treating the peptide-resin with 20% piperidine in dimethylformamide (DMF) for 3 min, followed by another cycle of 10 min and four times DMF washed. Amino acids were activated using DIC (*N,N*′-diisopropylcarbodiimide) and Oxyma pure at 0.5 mM in DMF. 

Peptides were cleaved from the resin with trifluoroacetic acid (TFA): triisopropylsilane solution (95:5 *v*/*v*), precipitated and washed with cold diethyl ether. Purification of crude peptides was carried out using reverse-phase HPLC with a semi-preparative C18 column (Symmetry 300, Waters, Milford, MA, USA) and mobile phases of 0.1% TFA in water (eluent A)/0.08% TFA in acetonitrile (eluent B). A linear gradient from 5% to 100% of eluent B was applied over 60 min. 

Peptide-containing peaks were collected and characterized by matrix-assisted laser desorption/ionization-time-of-flight (MALDI TOF/TOF) mass spectrometry using an ABI 4800 mass spectrometer (Applied Biosystems, Waltham, MA, USA), operating in reflector mode. Peptide solutions with a purity greater than 95% were frozen, dried, and the powders were stored at −20 °C until use.

### 4.3. Synthesis of Cyclic Peptides Using the Dithiol Bis-Alkylation Method

Cyclic peptides were obtained using the Dithiol Bis-Alkylation method, which inserts a linker between the cysteines present in C-terminal and N-terminal positions as reported in [32]. Purified linear peptides were dissolved in a solution of acetonitrile: water (1:1, *v*/*v*) and buffered at pH 8.0 with 20 mM ammonium bicarbonate at a concentration of 1 mM. The peptides were then incubated for two hours at room temperature with 1.5 equivalents of the linker (dibromo-m-xylene, my). The reaction was stopped by lowering the pH with TFA, and the peptides were purified using HPLC. Peptide cyclization was confirmed using MALDI TOF/TOF mass spectrometry, and the resulting cyclic peptide had a molecular weight of 102 Da, higher than the linear peptide, indicating successful cyclization. After purification, the peptides were lyophilized and stored at −20 °C until use.

### 4.4. RBD-ACE2 Binding Assay

The ability of peptides to inhibit the interaction between the Wuhan variant RBD-Fc and ACE2 was evaluated using a commercial ELISA kit (RayBiotech, Peachtree Corners, GA, USA) with some minor modifications. Briefly, we used 1 µM non-tagged Wuhan type RBD as a positive control and tested it along with peptides (up to 100 µM) in triplicate. These were pre-incubated for 60 min with the ACE2 receptor at room temperature. Following this, the RBD-Fc protein was added to the ACE2-inhibitor solution, initiating a 120-min incubation at room temperature. The final steps of the ELISA test, including the addition of the anti-Fc secondary antibody, were carried out according to the kit’s instructions. The amount of RBD-Fc bound in the presence of the ACE2 peptides was determined by measuring the sample’s absorbance at 450 nm using a spectrophotometer (Infinite M200, Tecan, Männedorf, Switzerland). The affinity of the peptides for the ACE2 receptor was calculated using GraphPad Prism analysis software (Version 9.5). 

### 4.5. Surface Plasmon Resonance (SPR) Studies

We utilized a ProteOn XPR36 Protein Interaction Array system (Bio-Rad Laboratories, Hercules, CA, USA) equipped with six parallel flow channels to immobilize up to six ligands on the same sensor chip. Using conventional amine coupling chemistry, we immobilized anti-FLAG antibodies (Merck Life Science S.r.l, Milan, Italy) on two lanes [33]. One lane served as a reference, while the second lane captured FLAG-tagged ACE2 (AdipoGen, San Diego, CA, USA) on the chip. The anti-FLAG antibodies flowed for 5 min at a concentration of 30 µg/mL in acetate buffer, pH 5.0, on GLC sensor chips (Bio-Rad Laboratories, Milan, Italy) pre-activated according to the manufacturer’s instructions, with the remaining activated groups blocked with ethanolamine at pH 8.0. 

To test the binding of each compound, we followed a specific protocol. Firstly, we immobilized FLAG-ACE2 on the sensor chip by flowing an anti-tag antibody in phosphate buffered saline containing 0.005% Tween 20 (PBST), pH 7.4, for 5 min, resulting in around 1000 Resonance Units (RU) of immobilization. Next, we rotated the flow channels to enable parallel testing of up to six compounds or different conditions. We injected five concentrations (ranging from 1 to 100 μM, in PBST pH 7.4) in parallel for 180 s, followed by a 600 s dissociation phase. We then removed the bound inhibitor/ACE2 complex and regenerated the surface by injecting 20 mM HCl for 18 s at a flow rate of 100 µL. To test a new compound, we injected FLAG-ACE2 again. The SPR binding assays were conducted at a flow rate of 30 μL/min at 25 °C. Before analyzing the sensorgrams using BioRad’s analysis software (Version 3.1.0.6), we normalized them to a baseline of zero and subtracted the reference surface signal (anti-FLAG antibody). 

### 4.6. Cells and Pseudotyped Virus

Human embryonic kidney cells stably expressing human receptor ACE2 (HEK293-ACE2) were generated as described [34]. Cells were cultured in Dulbecco’s modified Eagle Medium (DMEM; Gibco/Euroclone, Milano, Italy, #ECB7501L) supplemented with 10% heat-inactivated fetal bovine serum (FBS, Gibco #10270), L-glutamine (Gibco, #25030-024), non-essential amino acids (Gibco/Euroclone, #ECB3054D), penicillin/streptomycin (Corning, New York, NY, USA, #20-002-Cl), and puromycin (Genespin, Milan, Italy). Cells were maintained in T25 flasks at 37 °C in humidified 5% CO_2_ and routinely split every 3–4 days. All the pseudotyped viral particles were commercially purchased (VectorBuilder Inc., Chicago, IL, USA). 

### 4.7. Cell Viability

HEK293-ACE2 cells were seeded on 96-well plates (2 × 10^4^ cells/well) in a complete DMEM medium with 10% FBS. After incubating 24 h at 37 °C in a humidified 5% CO_2_ atmosphere, the cell supernatant was replaced with fresh medium containing peptides, which were initially prepared as a 5 mM stock solution in Milli-Q water, and then added to the medium at final concentrations ranging from 1–100 µM. Cells were treated in the same experimental conditions with SARS-CoV-2 spike RBD Recombinant Protein (3–1000 nM). Control cells were treated with equivalent Milli-Q water (Vehicle). Cells were incubated for 24 h at 37 °C in humidified 5% CO_2_; then, the medium was replaced with fresh one. After an additional incubation of 24 h, cells were treated for four h at 37 °C with 5 mg/mL 3-(4,5-dimethylthiazol-2-yl)-2,5-diphenyltetrazolium bromide (MTT) (Sigma Aldrich, St. Louis, MO, USA, #M56%%-1G) in 5 mM phosphate buffered saline (PBS), pH 7.4. The MTT was then carefully removed, and cells were resuspended in acidified isopropanol (0.04 M HCl). Cell viability was determined by measuring the absorbance at 560 nm using a spectrophotometer (Infinite M200, Tecan, Männedorf, Switzerland).

### 4.8. Transduction Assay

HEK293-ACE2 cells (2 × 10^4^/well) were seeded on 96-well plates in complete DMEM medium with 10% FBS and incubated overnight at 37 °C in humidified 5% CO_2_ atmosphere. The medium was then removed from the plate, and a fresh one containing 1–100 µM of cyclic or linear peptides was prepared as a 5 mM stock solution in Milli-Q water. Cells were treated under the same experimental conditions with SARS-CoV-2 spike RBD Recombinant Protein (3–100 nM) Wuhan (AcroBiosystem, Newark, DE, USA, SPD-C52H3) or Omicron (GenScript, Piscataway, NY, USA, Z03728) as the positive control or an equivalent volume of Milli-Q water as a negative control. Cells were incubated for four h at 37 °C in humidified 5% CO_2_ atmosphere and then infected with lentiviral vectors exposing the selective SARS-CoV-2 spike protein (5–50 MOI) in the presence of 10 µg/mL Polybrene (VectorBuilder). Pseudotyped lentivirus with no spike (bald) and lentivirus-infected non-peptide treated cells (vehicle) was used as controls. The day after the transduction, the medium was replaced with a fresh one. After an additional 24 h of incubation, the transduction efficiency was evaluated by determining the percentage of cells expressing GFP or RFP using a ZOE^TM^ fluorescent cell imager (Bio-Rad, Hercules, CA, USA). As previously described, ZOETM images were analyzed with the Fiji software (Version 2.9.0), an open-source platform for biological image analysis [34].

### 4.9. Statistical Analysis

Statistical analyses were performed using Prism GraphPad software v.9.4 (GraphPad Software, San Diego, CA, USA). All data points were included, except for experiments where negative and/or positive controls did not give the expected outcome. No test for outliers was performed. Data were analyzed using an unpaired *t*-test or one-way ANOVA corrected by a Bonferroni post hoc test, and the results were expressed as means ± SD. A *p*-value less than 0.05 was considered significant.

## 5. Conclusions

The results of our study present a promising technological platform for developing and validating SARS-CoV-2 inhibitors, crucial in pandemics like COVID-19. In addition to facilitating a rapid method for identifying SARS-CoV-2 inhibitors, our platform could potentially be extended to tackle other viruses.

## Figures and Tables

**Figure 1 ijms-24-12146-f001:**
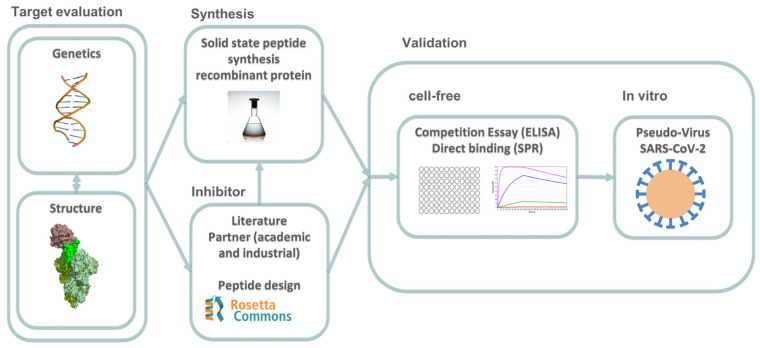
Flow chart of the drug discovery pipeline for designing and testing peptides inhibiting the SARS-CoV-2 infectivity. The pipeline involves target evaluation, inhibitor identification, and peptide synthesis, followed by cell-free and in vitro compound validation. Inhibitors can be designed using in silico tools or from the literature or external partners. Promising compounds are identified through ELISA and SPR assays before undergoing pseudovirus entry assay validation. This pipeline allows for the rapid identification of potential drug candidates for COVID-19.

**Figure 2 ijms-24-12146-f002:**
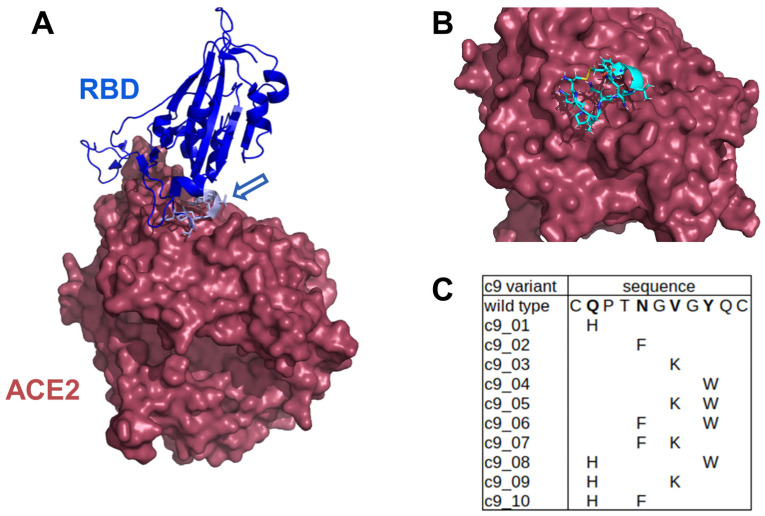
Identification of potential small peptide inhibitors using the Rosetta Peptiderive protocol. (**A**) Crystal structure of the RBD (blue) and ACE2 (brown) complex (PDB: 6M0J) with the identified cyclizable peptide shown in light blue and indicated by the arrow. (**B**) Surface representation of the cyclizable peptide bound to the ACE2 receptor. (**C**) List of potential cyclizable peptides, including the wild type and 10 mutants with single or double amino acid changes.

**Figure 3 ijms-24-12146-f003:**
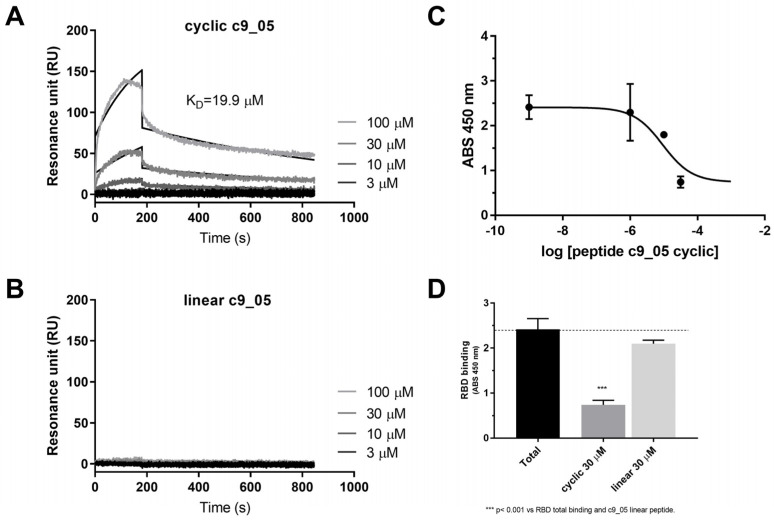
Direct binding and inhibitory effect of cyclic peptide c9_05. Binding signal of (**A**) cyclic and (**B**) linear c9_05 to ACE2 as measured by SPR, with a sensorgram for four different c9_05 concentrations (3, 10, 30, and 100 μM) and fitting results displayed in black. (**C**) Concentration-dependent reduction in binding of RBD when ACE2 was preincubated with cyclic c9_05 at different concentrations (10, 30, 100 µM), as measured by ELISA competition assay. (**D**) Binding reduction in RBD to ACE2 when 30 μM of cyclic or linear c9_05 are present in the ELISA competition assay. *** *p* < 0.001.

**Figure 4 ijms-24-12146-f004:**
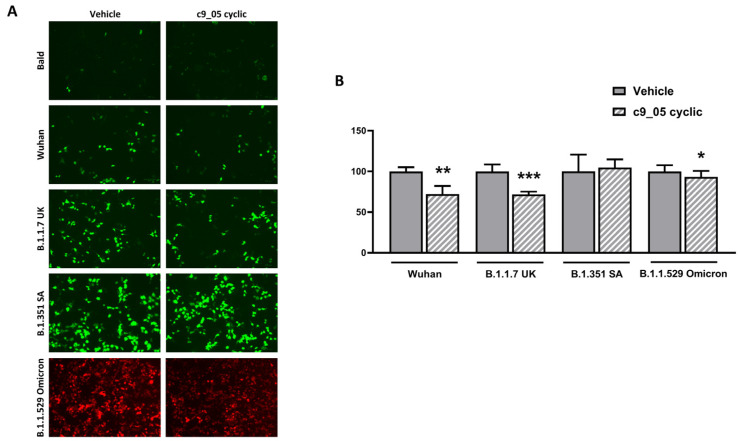
The treatment with cyclic c9_05 reduced pseudoviral transduction. (**A**) Representative fluorescence microscopy images and (**B**) percentage of GFP (green) or RFP (red)-transduced HEK293-ACE2 cells pre-incubated with 25 µM c9_05 cyclic peptide in water and then transduced with the pseudoviral particles exposing different SARS-CoV-2 spike isoforms or with pseudoviral particles without (bald). Cells were pre-incubated in the same experimental conditions, with an equivalent volume of Milli-Q water as a negative control (Vehicle). Cells were also pre-incubated with 25 µM c9_05 cyclic peptide in water but not transduced (c9_05 cyclic). (**A**) Scale bar, 100 µM. (**B**) Data are the mean ± SD of GFP-positive HEK293-ACE2 compared to control cells incubated with vehicle only. * *p* < 0.05, ** *p* < 0.01 and *** *p* < 0.001 vs. vehicle according to unpaired *t*-test.

## Data Availability

All data generated and analyzed during this study are included in the published article.

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
