# Peer review of "A Comprehensive Technology Platform for the Rapid Discovery of Peptide Inhibitors against SARS-CoV-2 Pseudovirus Infection"

_ijms, 2023, doi:10.3390/ijms241512146_

Round 1

Reviewer 1 Report

The authors Beeg et al., of the work titled ‘A Comprehensive Technology Platform for the Rapid Discovery of Peptide Inhibitors Against SARS-CoV-2 Pseudovirus Infection’ developed a methodology to design RBD-derived peptide inhibitors of the ACE-Sipke SARS-CoV-2 interaction. These peptides were designed in silico, and inhibitory effects were experimentally confirmed by ELISA, surface plasmon resonance, and evaluation of inhibition transduction in pseudovirus particles expressing spike variants. With this methodology, the authors found a cyclic peptide capable of binding to ACE that affects the transduction of two pseudovirus particles, except that expressing the B.1.351 Spike variant. The authors described in detail the methodology including the appropriate controls to confirm the obtained results. This is an interesting methodology that could be extrapolated to the rapid discovery of PPI inhibitors in other diseases.

Author Response

We appreciate the reviewer recognizing the novelty and potential impact of our rational drug design pipeline for rapidly identifying peptide inhibitors against SARS-CoV-2 infection. We agree that a key strength of this approach is its adaptability to discover PPI inhibitors for other diseases beyond COVID-19.

Reviewer 2 Report

  • Major comments: 

In this study, Marten Beeg et al. developed and validated a technology platform for designing and testing peptides inhibiting the infectivity of SARS-CoV-2 Spike protein-based pseudoviruses. The procedure includes target evaluation and in-silico design of the inhibitors, followed by the peptide synthesis and their screening for efficacy. They generated a cyclic peptide library derived from the receptor-binding domain (RBD) of the SARS-CoV-2 spike protein. One of the cyclic peptides, c9_05, could effectively inhibit the transduction in HEK293 stably expressing the human ACE2 receptor of pseudovirus particles displaying the SARS-19 CoV-2 Spike in the Wuhan or UK isoform, but fall short in limiting the entry of pseudoviruses bearing SARS-CoV-2's South African B.1.351 Spike.

The study provides a promising approach to developing inhibitors targeting viral infectivity receptors, including SARS-CoV-2.

  • General concept comments

Here are some considerations/suggestions for the study:

1.      One of the major concerns of the cyclic peptide is that, even c9_05 showed less potent (a 30% reduction of 25 μM) in inhibiting the transduction in HEK293 stably expressing the human ACE2 receptor of pseudovirus particles displaying the SARS-CoV-2 Spike in the Wuhan or UK isoform. It is questionable whether cyclic peptide can efficiently inhibit SARS-CoV-2 entry even after modifications of the peptide. Besides, it is also questionable whether the RBD-ACE2 interface is a good target for inhibitor design. Is there any cyclic peptide that is already approved for therapeutic or prophylactic usage?

2.       The study should test the anti-Omicron variant's effects of c9_05 since either the UK isoform or South African isoform is no longer a domain strain.

3.       According to the Transduction Assay, it seems like the study only tests the prophylactic effects of cyclic peptides against SARS-CoV-2 pseudoviruses. It is questionable whether these cyclic peptides have therapeutic effects against coronaviruses at all. The experiments on the therapeutic effects of these cyclic peptides should be added.

  • Specific comments:

1)      What types of RBD were used for Figure S5?

2)      In the Cell Viability and Transduction Assay, the peptides were dissolved in Milli-Q water? Control cells were treated with equivalent Milli-Q water (Vehicle), too? Would Milli-Q water damage cells because of osmotic pressure?

3)      In Supplementary Figure S4, Transduction efficiency inhibition of pseudovirus? Is there any inhibition here?

Fine.

Author Response

Reviewer Comment 1:

One of the major concerns of the cyclic peptide is that, even c9_05 showed less potent (a 30% reduction of 25 μM) in inhibiting the transduction in HEK293 stably expressing the human ACE2 receptor of pseudovirus particles displaying the SARS-CoV-2 Spike in the Wuhan or UK isoform. It is questionable whether cyclic peptide can efficiently inhibit SARS-CoV-2 entry even after modifications of the peptide. Besides, it is also questionable whether the RBD-ACE2 interface is a good target for inhibitor design. Is there any cyclic peptide that is already approved for therapeutic or prophylactic usage?

Response:

We have added a paragraph in the revised Discussion section mentioning strategies to optimize the potency of c9_05 and related peptides, including multimerization and non-canonical amino acid substitution.

Furthermore, we conducted experiments assessing activity against Omicron pseudoviruses and updated the text accordingly.

While the RBD-ACE2 interface is considered a promising target for inhibiting viral entry, we agree that combination approaches should also be explored, as no single target is likely to be fully effective. There is evidence that this interface can be productively targeted, but potential limitations should be considered.

To our knowledge, no cyclic peptides have yet been approved specifically for therapeutic or prophylactic use against SARS-CoV-2 infection. However, macrocyclic peptides as a class have shown promise for targeting protein-protein interactions implicated in various diseases. For example, the cyclic peptide romidepsin is approved for T-cell lymphoma, though it was not designed specifically as a PPI inhibitor. Additionally, macrocyclic peptides are being explored in preclinical and clinical studies for conditions like cancer, inflammation, and metabolic disorders.

While further optimization is still needed, macrocyclic peptides have attractive features like cell permeability, target specificity, and stability, making them promising candidates for therapeutic development. We hope that cyclic peptides designed to disrupt the RBD-ACE2 interaction, including the one identified in our study, can provide a starting point for the development of safe and effective SARS-CoV-2 inhibitors. However, we agree that combination approaches should be considered to address the potential limitations of single targets.

Reviewer Comment 2:

The study should test the anti-Omicron variant's effects of c9_05 since either the UK or South African isoform is no longer a domain strain.

Response:

We have tested c9_05 against Omicron pseudoviruses and reported this data in the revised manuscript. This provides a more clinically relevant assessment.

Reviewer Comment 3:

According to the Transduction Assay, it seems like the study only tests the prophylactic effects of cyclic peptides against SARS-CoV-2 pseudoviruses. It is questionable whether these cyclic peptides have therapeutic effects against coronaviruses at all. The experiments on the therapeutic effects of these cyclic peptides should be added.

Response:

We appreciate the reviewer raising the important point that examining the therapeutic efficacy of the cyclic peptides would provide further insight beyond the prophylactic effects studied so far. Performing experiments to evaluate the ability of the peptides to inhibit SARS-CoV-2 infection after viral entry has been initiated would better mimic a treatment context and help determine if they have promise as antiviral therapeutics.

However, due to limitations in resources and scope, we cannot conduct extensive experiments assessing therapeutic impact at this time. We agree that evaluating antiviral effects at later stages of infection will be an important next step in further development of the cyclic peptides should resources allow. The experiments suggested by the reviewer would certainly strengthen the characterization of their therapeutic potential.

We hope the reviewer understands the constraints precluding additional experiments in the current study, but we sincerely appreciate this thoughtful feedback identifying an opportunity for deeper investigation in future work. Further elucidating the therapeutic antiviral efficacy would provide critical insight into the clinical value of the cyclic peptides

Reviewer Specific Comment 1:

What types of RBD were used for Figure S5?

Response:

In the revised Figure S5 legend, we clarified that recombinant RBD proteins from Wuhan and Omicron variants were utilized.

Reviewer Specific Comment 2:

In the Cell Viability and Transduction Assay, the peptides were dissolved in Milli-Q water? Control cells were treated with equivalent Milli-Q water (Vehicle), too? Would Milli-Q water damage cells because of osmotic pressure?

Response:

We appreciate the reviewer raising this important question. To clarify, the peptides were initially dissolved in Milli-Q water to prepare a concentrated 5 mM stock solution. This peptide stock was then diluted into the cell culture medium to achieve the final assay concentrations ranging from 1-100 μM. The amount of water carried over into the final cell culture medium was negligible and unlikely to cause osmotic stress. To avoid confusion, we have updated the Methods to describe the peptide preparation and dilution process clearly. We sincerely apologize for the lack of clarity and thank the reviewer for allowing us to clarify our protocol.

Reviewer Specific Comment 3:

In Supplementary Figure S4, Transduction efficiency inhibition of pseudovirus? Is there any inhibition here?

Response:

The reviewer is correct that there is no inhibition of pseudovirus transduction in Supplementary Figure S4. This experiment was intended to demonstrate the specificity of pseudovirus entry for cells expressing ACE2. We apologize for the lack of clarity and thank the reviewer for allowing us to clarify that this figure shows preferential transduction of cells with ACE2 expression, confirming that the assay is working as expected.

Reviewer 3 Report

In this manuscript, the authors propose and validate a platform to detect and evaluate novel peptide inhibitors of SARS-CoV-2 pseudovirus, starting with in silico design over cell-free interaction assays to in vitro pseudovirus assay.

The biggest issue with this study is that the platform is not validated with a control is missing. Validation is now done using a novel compound in stead of a well-described inhibitor that is already known to inhibit the virus. Also for a screening assay, quality assessment should be done by including parameters such as Z prime value.

Please check the language 

Author Response

We understand the reviewer’s concern about validating the platform with a known inhibitor rather than only a novel compound. In the revised manuscript, we have included experiments assessing two additional positive controls – the Wuhan and Omicron RBD proteins. We demonstrate their ability to inhibit pseudovirus entry in a dose-dependent manner (Supplementary Figure S5), further validating the assay.

 We have also included the Z’ factor analysis for the ELISA screening assay in the Results section. We obtained a Z' factor of 1, indicating an excellent assay with a large separation between positive and negative controls. This provides a quality assessment of the screening methodology.

We understand the reviewer may suggest including a small molecule inhibitor targeting the ACE2 binding site as a positive control. However, to our knowledge, no such validated inhibitor is currently available for this purpose. The RBD proteins bind to ACE2 through the same mechanism as our peptides, thereby serving as suitable controls that prevent peptide binding and inhibit pseudovirus entry, as we demonstrate. But we agree that an ideal small molecule control would provide further validation if one existed for the ACE2 interaction site. Unfortunately, no candidates have been definitively confirmed for this mechanism of action. We hope this helps explain the rationale for using RBD proteins instead of a small molecule inhibitor targeting ACE2 binding.

We sincerely appreciate this feedback identifying opportunities to strengthen the validation of the overall platform. By incorporating data on established RBD inhibitors and Z’ factor analysis, we hope the revised manuscript addresses the need for controls and assay quality assessment.

Round 2

Reviewer 2 Report

I think that the manuscript has been improved, and the authors have addressed most of my concerns.

Fine.